# Nitrous Oxide Abuse: Clinical Outcomes, Pharmacology, Pharmacokinetics, Toxicity and Impact on Metabolism

**DOI:** 10.3390/toxics11120962

**Published:** 2023-11-28

**Authors:** Emeline Gernez, Graham Robert Lee, Jean-Paul Niguet, Farid Zerimech, Anas Bennis, Guillaume Grzych

**Affiliations:** 1CHU de Lille, Centre de Biologie Pathologie Génétique, 59000 Lille, France; emeline.gernez@chu-lille.fr (E.G.); farid.zerimech@chu-lille.fr (F.Z.); 2Mater Misericordiae University Hospital, D07 R2WY Dublin, Ireland; glee@mater.ie; 3Service de Neurologie, Hôpital Saint Vincent de Paul–GHICL, 59000 Lille, France; niguet.jeanpaul@ghicl.net; 4Assistance Publique—Hôpitaux de Paris, Service de Neurologie, Groupe Hospitalier Universitaire Paris Sud, Hôpital Bicêtre, 94270 Le Kremlin-Bicêtre, France; anas.bennis@aphp.fr

**Keywords:** nitrous oxide, cobalamin, homocysteine, methylmalonic acid, vitamin, neurology, toxicity, addiction, vitamin B12, neuropathy

## Abstract

The recreational use of nitrous oxide (N_2_O), also called laughing gas, has increased significantly in recent years. In 2022, the European Monitoring Centre for Drugs and Drug Addiction (EMCDDA) recognized it as one of the most prevalent psychoactive substances used in Europe. Chronic nitrous oxide (N_2_O) exposure can lead to various clinical manifestations. The most frequent symptoms are neurological (sensitive or motor disorders), but there are also other manifestations like psychiatric manifestations or cardiovascular disorders (thrombosis events). N_2_O also affects various neurotransmitter systems, leading to its anesthetic, analgesic, anxiolytic and antidepressant properties. N_2_O is very challenging to measure in biological matrices. Thus, in cases of N_2_O intoxication, indirect biomarkers such as vitamin B12, plasma homocysteine and plasma MMA should be explored for diagnosis and assessment. Others markers, like oxidative stress markers, could be promising but need to be further investigated.

## 1. Introduction

The recreational use of nitrous oxide (N_2_O), also called laughing gas, has increased significantly in recent years. In 2022, the European Monitoring Centre for Drugs and Drug Addiction (EMCDDA) recognized it as one of the most prevalent psychoactive substance used in Europe [1]. N_2_O is typically consumed by inhalation from balloons for its euphoric properties, which last only a few minutes. However, this consumption is associated with a risk of both acute and chronic toxicity. Chronic toxicity is rising in recent years because of an increase in consumption rates (in terms of quantity and frequency) related to the dependence and the tolerance effect of this molecule. There are currently no official recommendations for clinico-biological management, but a European group was set up this year to establish guidelines for this public health issue: https://www.eflm.eu/site/page/last/1832 (accessed on 27 November 2023). The aim of this review is to provide an update on the clinical manifestations, pharmacological effects and biological effects of N_2_O intoxication. A comprehensive literature review was conducted, sourcing relevant articles from pubmed and medline databases to provide a thorough examination of the current state of knowledge on the subject.

## 2. Clinical Manifestations

### 2.1. Brief History of the First Reported Manifestations of Chronic N_2_O Exposure

The first adverse effects of N_2_O, reported in 1952, were hematological in nature described in young patients infected with tetanus to relieve their pain. One of the patients, a 15-year-old boy, died days later from septicemia and granulocytopenia secondary to severe bone marrow depression [2]. Several years later, one of the authors coined this incident as the time “when nitrous oxide lost its innocence” [3].

The first neurological adverse effects were reported in 1978 by Robert Layzer in San Francisco, even without Magnetic Resonance Imaging (MRI) and biological background [4,5]. Many case reports have since been described and the number of patients drastically increasing over time. A small increase in publications and cases were reported in the late 90′s due to an increased recreational use of N_2_O among dentists and healthcare givers in the USA who had easy access to the gas [6]. The use of whipped cream cartridges become a new phenomenon and was described in many countries as “nanging” [7].

### 2.2. Common Symptoms and Signs

As reported initially by Layzer, patients had a mixture of central and peripheral nervous system symptoms (Figure 1). In several examinations, subjects even alternated between hyper and hyporeflexia, as myelopathic or neuropathic influences predominated. This presentation was called “myeloneuropathy” [4]. Before MRI, Lhermitte sign and sphincter disturbances were suggestive of myelopathy. Some patients were thought to have multiple sclerosis because of a relapsing course punctuated by resumed consumption.

Chronic N_2_O users have common mild neuropathic symptoms. A large survey conducted with more than 240,000 people concluded that 17% of participants indicated a chronic use of N_2_O, with 4.2% of recreational users reporting persistent paresthesia or numbness, distributed in a stocking glove pattern, suggestive of a length-dependent axonal polyneuropathy, as seen in many others toxic neuropathies [8].

Moreover, N_2_O seems to have an association with motor deficit in the lower limbs and proprioceptive ataxia. In a recent French study, the authors described symptoms and used scores to quantify the severity of patients’ symptoms [9]. At first evaluation, median lower limb Medical Research Council (MRC) sum score was 26 (normal: 30), median Romberg score was 2 (meaning significant imbalance without falls on Romberg maneuver; from 0 normal to 3 fall), median modified INCAT Sensory Sumscore was 3.5 (normal: 0, maximum score: 16). Another recent study found that in patients with N_2_O-induced neuropathy, at least half of the MRC sum score reduction was due to anterior tibialis weakness in 60.4% of patients [10]. Additionally, in a Taiwanese study from 2019, authors compared N_2_O abusers and vitamin B12 deficiency patients, and found that the former had more severe motor deficit quantified by the MRC and the Neuropathy impairment Score, predominantly in the lower limbs [11].

Among N_2_O users, an Australian team hypothesized that patients may have a more severe presentation according to their vitamin B12 levels, but did not find any significant differences between the two groups [12]. A recent study showed that plasma methylmalonic acid and plasma methionine levels were significantly correlated with the clinical severity. Homocysteine was also correlated with the severity of presentation, but the magnitude of effect was less prominent. Vitamin B12 was not correlated to the clinical severity [13,14].

However, since neurological examination seem to not reliably distinguish the central and peripheral presentations, we will try to discuss different system involvements according to paraclinical results.

### 2.3. Central Nervous System Involvement

The classical spinal magnetic resonance presentation is a hypersignal on sagittal T2-weighted images in the cervical level, and to a lesser extent, in the thoracic level. An inverted “v” shaped signal is observed on the axial plane, corresponding to the dorsal columns. Interestingly, these lesions are rarely responsible for a spinal sensory level on examination. In a Chinese study describing 15 patients with spinal cord lesions, 80% had a cervical lesion and 86.7% did not have a spinal sensory level on clinical examination. This may be explained by their non-transverse nature. The vulnerability of the cervical segment had been attributed to the higher density of myelinated fibers of the dorsal columns in the cervical segment compared with the thoracic segment [15]. This study also demonstrated that these lesions were longitudinally extensive, affecting 5 to 6 vertebral levels in most patients [15]. This is of particular interest because it can mimic other inflammatory spinal cord diseases, like neuromyelitis optics spectrum disorder [16].

However, a large group of patients have a normal spinal MRI. Two studies demonstrated that patients who had a long exposure time to N_2_O (i.e., >6 months) had a greater probability of having a normal MRI [17,18]. It is important to note that this does not mean that these patients do not have a myelopathy. This may be explained by the regression of edema secondary to demyelination when the exposure is more recent. A Chinese study investigated differences between an Asian cohort and multiple non-Asian patients from the literature. They found that non-Asian patients were significantly younger, with a sex ratio more shifted towards males, had more history of multiple drug abuse, had a longer duration of exposure to N_2_O before developing symptoms, and were more likely to develop an isolated myelopathy than a combined injury. The authors hypothesized that MTHFR genotype status was a possible explanation because the frequency of pathologic alleles differ from one population to another [15].

Finally, N_2_O abusers seem to have more frequent spinal cord involvement than patients with vitamin B12 deficiency [11].

### 2.4. Peripheral Nervous System Involvement

In the first report from Layzer in 1978, the electromyographic (EMG) examination showed normal to subnormal sensory conduction studies, with predominantly abnormal motor conduction studies. It was concluded that neuropathy was due to axonal degeneration rather than to segmental demyelination. One patient had a sural-nerve biopsy that showed only non-specific axonal degeneration [5].

Many N_2_O abusers have persistent paresthesia or numbness distributed in a stocking-glove pattern, suggesting a length-dependent axonal polyneuropathy. Patients have predominantly lower limb motor axonal injury, consisting of a motor length-dependent axonal polyneuropathy, sometimes associated with demyelinating injury in the upper limbs [8]. From an electrophysiological perspective [19], there seem to be specific features not usually seen in classic length-dependent polyneuropathies, such as:More motor and sensory nerve injury in the lower limbs compared to the upper limbs,More motor nerve injury than sensory nerve injury in the lower limbs,More demyelinating features in the sensory and motor nerves of the upper limbs, with a marked motor predominance.

The authors also investigated factors influencing the severity of motor nerve axonal injury, and found that a longer exposure time and disease course were significantly associated with more severe motor axonal injury in the lower limbs [19].

When compared to other published studies, data from the literature suggest a variable rate of mechanism of neuropathy. Authors report mainly axonal or mixed axonal and demyelinating neuropathies, with a nearly constant rate of demyelinating neuropathies of between 5 and 8% [19,20,21,22]. We believe this variability is probably due to differences in the definition of demyelination on EMG.

Moreover, a large number of these patients can present with an acute onset neuropathy mimicking Guillain Barré syndrome. Many of these patients were treated inappropriately with intravenous immunoglobulins [23]. To avoid unnecessary treatment, some authors investigated clinical or electrophysiological clues, and established a list of elements to differentiate the two conditions (Table 1) [10,24].

Finally, N_2_O abusers seem to have prominent motor super excitability changes and less prominent sensory super excitability changes, suggesting a unique pathological process affecting the paranodal region [11].

To summarize, N_2_O consumption seems to give peculiar clinical pictures that can be subdivided to three presentations (Figure 1):Chronic sensory length-dependent axonal polyneuropathy,Subacute motor and/or ataxic predominant length-dependent axonal polyneuropathy,Acute sensory ataxic and/or motor axonal polyradiculoneuropathy.

We suppose that nitrous oxide is responsible for predominantly motor neuropathies because it specifically targets the node and paranode region, giving a distinct phenotype of neuropathies, very different from what is seen in patients with vitamin B12 deficiency. A subgroup of patients also seem to have a demyelinating pattern on EMG.

### 2.5. Prognosis of Central and Neurological Nervous System Involvement

Data regarding the long-term prognosis of neurological consequences of N_2_O consumption is scarce. Many patients do not come back to their follow up medical appointment. A French study reported the evolution of 6 patients after a mean time of 4.9 months (interquartile range 3.0–6.25). They evaluated their Overall Neuropathy Limitations Scale, which measures disability due to peripheral neuropathy, and found that they have a score of 2 on the lower limbs on second evaluation, corresponding to an independent but abnormal looking gait [9]. In a systematic review of the literature cases, among 59 patients for which information at follow up was available, symptoms completely resolved in only 17%, and improved partially in 78%. In 5% of patients, there was no improvement at all [25]. Prognostic factors were specifically investigated in a large cohort of N_2_O abusers in China. The authors found that only female sex was independently associated with better outcomes. However, they did not define how they determined a complete or partial recovery, and they did not include relevant variables such as methylmalonic acid [22].

### 2.6. Other Presentations

Regarding the cognitive aspect, vitamin B12 deficiency is a known etiology of dementia-like clinical presentation that can improve after supplementation [26]. A French study conducted in rats concluded with a diminished exploration ratio when the animals had been exposed to N_2_O in a dose-dependent manner. Their performances become better with time, suggesting a working memory impairment [27]. However, we did not find any study that systematically explored this aspect in N_2_O abusers.

Patients may also exhibit psychiatric symptoms due to chronic N_2_O use (Figure 1). A review of all case reports published in the literature found that psychotic symptoms and sleep disturbance were the most frequent symptoms in these patients [28]. There is also an association between chronic N_2_O use and anxiety and depression, as demonstrated in another Chinese study, where higher scores on Hamilton anxiety and depression scales were significantly associated with clinical severity scores and homocysteine levels [18]. However, it is important to mention that individuals experiencing anxiety and depression might be more inclined to use N_2_O as a means of alleviating symptoms; thus, the causality is difficult to establish.

Finally, there have been two case reports regarding thromboembolic cerebral presentations. A French team reported the case of a patient who presented with a cerebral venous thrombosis in the context of N_2_O abuse [29]. The extensive workup for thrombophilia did not show any alternative cause, including mutations of the MTHFR gene. To note, deep venous thrombosis and pulmonary embolism are now well established complications of chronic N_2_O use, but cerebral venous thrombosis seem to occur rarely in this setting [30]. The second case describes a 32-year-old patient presenting with a right middle cerebral artery ischemic stroke secondary to a non-occlusive thrombus of the internal carotid artery [31]. He had a 5-year-long history of recreational use of N_2_O, and had elevated levels of homocysteine. However, the authors did not rule out the possibility of a cervical artery dissection.

## 3. Pharmacological Effects

The reported pharmacological effects of N_2_O depend on its concentration. For example, in aesthesia where it is used in an equimolar mixture with O_2_ (EMOMNO), a concentration of 25% N20 is considered adequate for pain reduction, with hypnotic effects above 60% and unconsciousness at 70% [1]. Some effects from the recreational use of N20 arise from hypoxia, caused by inhaling the gas and the displacement of oxygen. In one study mimicking such misuse and delivery, bolus administration of 80% N20 showed psychomotor impairment as early as one minute after inhalation [32]. There is usually complete recovery from its main effects within a few minutes of exposure, although delayed or lingering behavioral effects may persist for up to 30 min [1].

### 3.1. Dependence Producing Potential of Nitrous Oxide

This short-lived psychoactive effect of N_2_O may precipitate frequent and heavy use. Coupled with this is N_2_O’s reinforcing effect [33,34,35], which is the ability of a drug to increase the probability that it will be self-administered again. The nucleus accumbens (NAcc), located in the basal forebrain, is involved in motivation, pleasure and reward. In the NAcc, antagonism of the NMDA (N-methyl-D-aspartate) subtype of glutamate receptors by N_2_O causes disruption of glutamate homeostasis, which otherwise helps establish and maintain drug-seeking behavior [36]. Genetic variability at the receptor level is a potential factor affecting susceptibility to developing N_2_O dependence [37]. Considering that administration of NMDA antagonists may reduce drug-seeking behavior, this may suggest the further therapeutic potential of N_2_O in the treatment of substance use disorder [38]. However, nitrous oxide’s potential to modulate other neurotransmitter systems, including stimulation of the dopaminergic system [1], involved in reward and motivation may support the overriding reinforcing effect of N_2_O and addiction. The latter was subject of earlier review by Gillman [39] almost 30 years ago and more recently by Back et al. [40]. Although there appears to be no consensus on its addictive potential, the literature does report N_2_O as meeting many of the substance use disorder (SUD) symptoms; and with recently reported high incidence of the same [41], we concur that it is reasonable amidst the paucity of (pre-) clinical evidence to consider N_2_O as potentially addictive.

N_2_O is recognized for its anesthetic, analgesic, anxiolytic and anti-depressant effects [1,42] and the molecular targets of N_2_O and concomitant regulation of key neurotransmitters (glutamate, opioid, noradrenaline and γ-aminobutyric acid (GABA) will be considered in turn with major neuropharmacological effects (Table 2, Figure 2).

### 3.2. Anaesthesia

N_2_O administration is via inhalation utilizing a simple face mask, laryngeal mask airway or an endotracheal tube. In accordance with the European Society of Anesthesiology Task Force on Nitrous Oxide, N_2_O is used at lower concentrations (30 to 50% with oxygen) for sedation in surgical and dental procedures and up 70% for general anesthesia with associated unconsciousness and immobility [43]. N_2_O is the least potent inhalational anesthetic, as defined by the minimum alveolar concentration (MAC), which prevents a movement response (immobility) during a painful (e.g., surgical) stimulus. Currently, non-competitive inhibition of the NMDA receptors, specifically the AMPA (α-amino-3-hydroxy-5-methyl-4-isoxazolepropionic acid) and kainite forms, is considered the main molecular target for N_2_O’s anesthetic effect [44]. These receptors comprise ligand-gated ion channels that are activated by glutamate. This neurotransmitter mediates the majority of excitatory synaptic transmission throughout the CNS and mediates the transmission of nociceptive messages and hyperalgesia as part of the glutamatergic system [45]. N_2_O thereby provides anti-nociception, blocking the detection of painful or injurious stimulus by sensory neurons, as well as analgesia. Other anesthetic effects include decrease in memory, perception, arousal, muscle tone and autonomic functions, in part attributed to nicotinic acetyl choline receptor inhibition [42]. Further possible molecular targets by which N_2_O may decrease excitability and slow down the transmission of electrical impulses include the potassium channels (inhibition) in the brain and spinal cord [46].

### 3.3. Analgesia

Analgesia is defined as insensibility to pain without loss of consciousness, and a property of general anesthesia [43]. The analgesic and anti-nociceptive effect of N_2_O involves the opioidergic system, by antagonism of the kappa opioid receptor, and the subsequent regulation of GABAergic and noradrenergic systems. In the periaqueductal grey (PAG) area of the midbrain, which is responsible for modulation of descending pain, blockade of these opioid receptors ablates nitrous-oxide-mediated analgesia, itself also partially reversed by the opioid receptor antagonist naloxone [47]. Corticotrophin releasing factor from the hypothalamus is also released in response to N_2_O [48] and causes activation of opiodergic neurons in the PAG with release of endogenous opioids such as dynorphins, which also activate kappa opioid receptors [49]. Concomitant blockade of inhibitory GABAergic neurons [50] in the pons region of the brainstem causes stimulation of pontine noradrenergic neurons in descending pathways to the spinal cord (dorsal horn), where α1-adrenoceptors on inhibitory GABAergic neurons and α2B adrenoceptors on ascending second-order neurons are activated with noradrenaline release. This effect causes decreased firing of the second-order neuron, hence leading to anti-nociception by reducing pain impulses ascending into the supraspinal regions [15].

### 3.4. Anxiolytic Effect

Anxiolytics are used to prevent or treat anxiety symptoms or disorders and include the benzodiazepine class of drugs. The anxiolytic effect of N_2_O involves activation of the gamma-aminobutyric acid type A (GABAA) receptor through its benzodiazepine binding site, though a direct effect is uncertain [51,52]. However, any such effect is considered minimal compared to the effect on NMDA receptors [53].

### 3.5. Anti-depressant Effect

N_2_O’s purported anti-depressant effect [54], which is a comparatively more recent and ongoing area of exploration, is mediated through non-competitive inhibition of NMDA receptors, and is considered analogous to that of ketamine and similarly short-lived [54,55]. The latter property perhaps hinders its use clinically as an anti-depressant. Other purported molecular targets and effects include the regulation of Brain-derived neurotrophic factor (BDNF) which has a role in synaptic plasticity, synaptogenesis and neurogenesis [56,57], contributing to its anti-depressant effect, as opposed to neuronal atrophy and synaptic loss, as seen with stress and depression.

The pharmacological impacts of N_2_O are resumed in Figure 2.

## 4. Laboratory Medicine

### 4.1. Direct N_2_O Measurement

N_2_O may affect driving behavior and may cause fatal car accidents. As such, detection is an important issue. N_2_O has a very short half-life of a few minutes [58], as the uptake and elimination curves are comparable [59]. N_2_O elimination is mainly pulmonary. When exposure ends, exhaled air concentration declines rapidly, from 66–70% to 6–9% at 5 min and to 2–4% at 30 min during normoventilation. The elimination is slower in cases of hypoventilation [60]. Thus, the measurement of N_2_O in exhaled air is not routinely usable for patients presenting to the emergency department due to the time gap between consumption and admission. The issue is the same for toxicology screening: indeed, for police roadside controls, this appears to be difficult due to the time gap between arrest and sample collection. A small fraction of N_2_O is excreted in urine in an unchanged form, but there are few pharmacokinetic data in the literature; the detection period remains unknown. Moreover, urinary N_2_O can also be produced by bacteria in the case of a urinary tract infection, so its presence is not specific to nitrous oxide exposure [59].

Additionally, there are technical difficulties concerning N_2_O measurement in biological fluids. First, gas chromatography–mass spectrometry (GC-MS) can be used, but has limitations, including the challenge of finding an optimal internal standard, the lack of sensitivity and the potential risk of leaks during sampling, extraction and analysis. Headspace-GC-MS, which is a method in which the sample is placed in a hermetically-sealed, gas-tight container, could be promising but need further studies to be used in laboratory medicine [61]. Infrared Spectroscopy techniques are sensitive methods to measure N_2_O in air, but not on biological matrices [62].

Consequently, N_2_O is not routinely measurable directly in biological matrices due to its kinetics and measurement issues. Thus, indirect markers related to the impact of N_2_O should be used.

### 4.2. Impact on Metabolism

N_2_O reacts with cobalamin, causing disruptions in One Carbon Metabolism, notably hyperhomocysteinemia. Both N_2_O and hyperhomocysteinemia increase oxidative stress.

#### 4.2.1. Cobalamin and One Carbon Metabolism

The clinical presentation of N_2_O intoxication is related to the functional impairment of vitamin B12, also called cobalamin (Figure 3). Indeed, N_2_O is a powerful oxidant agent: it leads to the oxidation of the cobalt ion of cobalamin(I) [63], resulting in the formation of cobalamin(II), unable to accept methyl groups. This results in a decrease in the formation of methylcobalamin, which is a cofactor for methionine synthase (MS or MTR). Consequently, there is a reduction in MS activity, an enzyme responsible for the conversion of homocysteine into methionine (involved in the synthesis of myelin gain) [64]. Cobalamin(II) also serves as a cofactor for methionine synthase reductase (MTTR) [65]. Therefore, we suggest that the accumulation of cobalamin(II) stimulates increased MTRR activity, which converts cobalamin(II) into cobalamin(III) through S-adenosylmethionine, subsequently amplifying homocysteine production. Very high levels of homocysteine are thus found in N_2_O-consuming patients, related to the two previous mechanisms. Major hyperhomocysteinemia in these patients can lead to thromboembolism events [66,67] because homocysteine can have an impact on different steps of coagulation pathways [68,69,70].

The influence of N_2_O on MMA-CoA mutase (Figure 3), which converts methylmalonic acid (MMA) into succinyl-CoA, is still debated [64]. Indeed, the oxidation of cobalamin(I) could secondarily result in a global cobalamin deficiency, including adenosylcobalamin deficiency, a cofactor for MMA-CoA mutase.

#### 4.2.2. N_2_O and Oxidative Stress

N_2_O has powerful oxidant properties. A study was conducted on 36 nurses occupationally exposed to anesthetics including N_2_O during surgical procedures. Biological assessments revealed an increase in oxidative stress markers, including thiobarbituric acid-reactive substances (TBARS) and F2 isoprostanes. There was also a significant decrease in the activity of the antioxidant enzyme glutathione peroxidase (GPX), and an increase in the levels of reactive oxygen species (ROS) in peripheral blood leukocytes [71]. Oxidative stress could partly explain the neurological impairment observed in N_2_O-consuming patients. Furthermore, some drugs target oxidative stress to combat certain neurodegenerative and neuroinflammatory effects [72,73].

#### 4.2.3. Homocysteine and Oxidative Stress

Hyperhomocysteinemia also enhances oxidative stress. In a study conducted on CBS (cystathionine beta-synthase) deficient mice [74], an inherited metabolic disease inducing severe hyperhomocysteinemia, several indicators of oxidative stress were notably increased. Lipid peroxidation markers, such as malondialdehyde (MDA) and 4-hydroxynonenal (HNE), were elevated, as well as protein-associated carbonyl groups, indicating protein oxidation. Another study on liver-specific, CBS-deficient mice [75] showed reduced levels of glutathione (GSH) and an altered systemic reactive oxygen species (ROS) status.

### 4.3. Indirect Biomarkers of N_2_O Intoxication

#### 4.3.1. Vitamin B12

As N_2_O leads to a functional vitamin B12 deficiency, the quantitative deficiency in vitamin B12 is secondary and inconsistent [14]. Patients are also frequently supplemented with vitamin B12; increased levels of vitamin B12 can also be found in intoxicated patients. Consequently, it seems more pertinent to investigate functional markers of vitamin B12 in cases of N_2_O intoxication, which are plasma MMA and plasma homocysteine.

#### 4.3.2. Plasma Homocysteine

Plasma homocysteine is highly sensitive and can be used as a marker of recent N_2_O consumption [14] because it rapidly increases in case of consumption. However, homocysteine levels decrease rapidly and can return to physiological values within several days after the last N_2_O consumption [76]. This biomarker is also not specific to N_2_O intoxication: plasma homocysteine increases in case of vitamin deficiency (vitamin B6, vitamin B9, vitamin B12), renal or hepatic injury, hypothyroidism and in certain metabolic diseases. Although not specific, homocysteine levels are often higher than those observed in metabolic diseases or deficiency states.

#### 4.3.3. Plasma MMA

Plasma MMA is more specific than homocysteine in the exploration of vitamin B12 status because it does not depend of vitamin B6 and B9 status; but rise in case of renal insufficiency and in certain metabolic diseases. However, it is not a sensitive marker of N_2_O abuse as its elevation is not consistent. Plasma MMA is correlated to the clinical severity [14]; thus, it can be used as a marker of clinical severity of N_2_O intoxication.

#### 4.3.4. Plasma Methionine

Methionine has also been investigated as a potential biomarker in nitrous oxide intoxication. Indeed, the decrease in MS activity could lead to a decrease in production of methionine, which is involved in the formation of myelin. The study made on 93 N_2_O consumers [13] showed a correlation between plasma methionine levels and clinical severity but observed no quantitative methionine deficiency. Therefore, even though severely affected subjects tend to have lower average methionine levels, they are within physiological values. The homocysteine/methionine ratio is positively correlated with clinical severity and can be used as an alternative to plasma MMA when it is not available.

#### 4.3.5. Oxidative Stress Markers

Oxidative stress markers could be of interest in N_2_O intoxication. However, no studies have been conducted on patients with a recreational use but only on occupational exposure. Therefore, additional investigations are needed to determine whether these markers may be of interest as a consumption marker or as a marker of clinical severity.

#### 4.3.6. Others Biological Parameters to Consider

Some biological parameters are crucial for the differential diagnosis of hyperhomocysteinemia, as this parameter is not specific of N_2_O intoxication. Thus, renal (creatinine) and hepatic exploration (AST, ALT, alkaline phosphatase, GGT) should be performed, as well as vitamin assessment (vitamin B6, vitamin B9), to explore nutritional deficiencies [77]. Cell blood count can be performed to investigate a potential anemia, although N_2_O does not appear to cause macrocytic anemia [23].

## 5. Conclusions

The recreational use of N_2_O has seen a significant increase in recent years, leading to a growing concern about its acute and chronic toxicity. There is a wide range of chronic manifestations including myelopathy, neuropathy, psychiatric manifestations, cognitive symptoms and cardiovascular effects. N_2_O interacts with neurotransmitter systems, leading to anesthetic, analgesic, anxiolytic and potential anti-depressant effects, with a potential dependance. Laboratory medicine plays a critical role in assessing N_2_O intoxication, with biomarkers such as plasma homocysteine, a marker of recent consumption, and plasma MMA, a marker of clinical gravity. Other biomarkers, like oxidative stress markers, could be interesting but need further investigations.

## Figures and Tables

**Figure 1 toxics-11-00962-f001:**
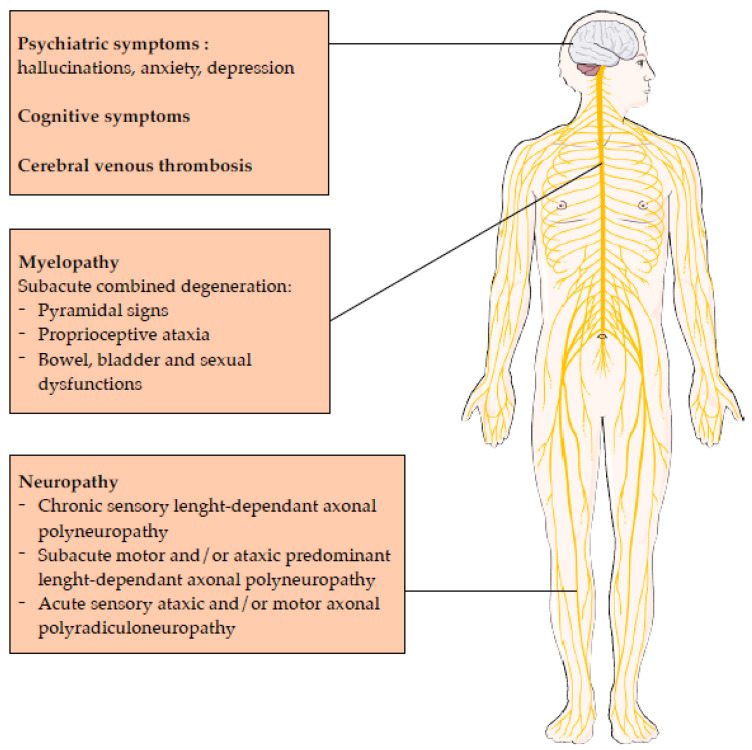
Clinical manifestations related to nitrous oxide intoxication.

**Figure 2 toxics-11-00962-f002:**
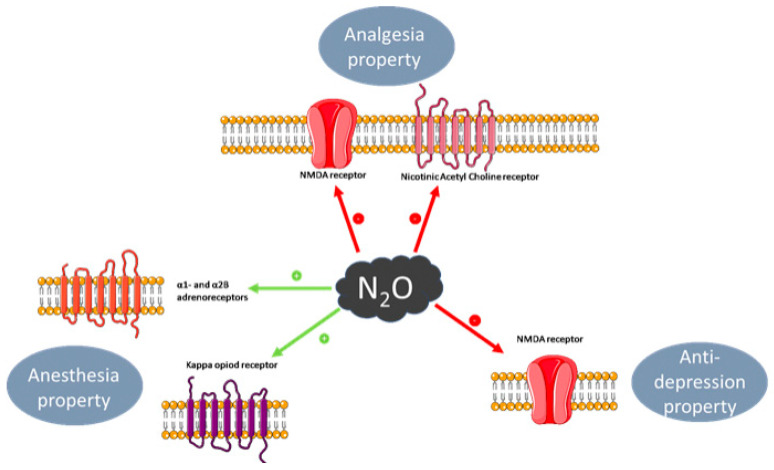
Main neurological effects of Nitrous Oxide, neurotransmitter modulation and receptor targets. Red arrow for inhibition and green arrow for activation.

**Figure 3 toxics-11-00962-f003:**
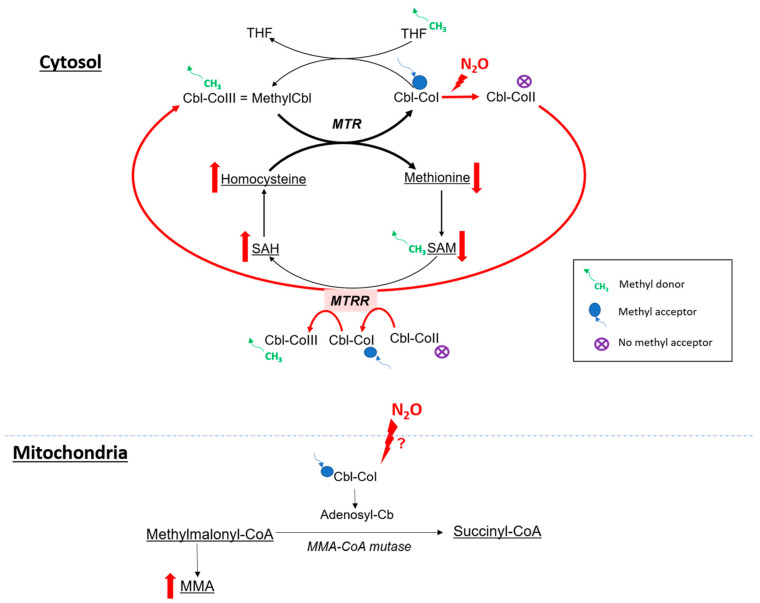
Hypothetic impact of N_2_O on metabolism. SAH: S-adenosyl-homocysteine, SAM: S-adenosyl-methionine, MTR: methionine synthase, MTRR: methionine synthase reductase, THF: tetrahydrofolate, MMA: methylmalonic acid. Red arrow for major pathway in case of cobalamin oxidation.

**Table 1 toxics-11-00962-t001:** Clinical and electrophysiological elements distinguishing Guillain-Barré syndrome from N_2_O-induced neuropathy.

	N_2_O-Induced Neuropathy	Guillain-Barré Syndrome
**Clinical characteristics**		
Upper limb weakness	Less frequent	More frequent
Numbness in the limbs	Common	Less frequent
Babinski sign	Common	Absent
Diffuse hyporeflexia/areflexia	Less frequent	Common
Ataxia	Frequent	Less frequent
Apallesthaesia	Common	Less frequent
Cranial nerve involvement	Uncommon	Frequent
Respiratory distress	Uncommon	Frequent
Swallowing problems	Absent	Frequent
**Electrophysiological characteristics**		
Conduction blocks	Uncommon	Frequent
Decrease in CMAP amplitude (upper limbs)	Less frequent	Frequent
Decrease in CMAP amplitude (lower limbs)	Frequent	Less frequent
Decrease in SNAP amplitude (lower limbs)	Frequent	Absent
Decrease in SCV	Possible	Uncommon
Absent F waves	Frequent	Less frequent

CMAP: compound muscle action potential; SCV: sensory conduction velocity; SNAP: sensory nerve action potential.

**Table 2 toxics-11-00962-t002:** Main neuropharmacological effects of nitrous oxide, neurotransmitter modulation and receptor targets.

Property	System	Receptor Target
Anesthesia	Glutamatergic inhibition	NMDA receptor (AMPA and kaitine) inhibition
	Cholinergic inhibition	Nicotinic Acetyl Choline receptor inhibition
Analgesia	Opioidergic activation (+GABAergic inhibition)	Kappa opioid receptor activation
	Noradrenergic activation	α1- and α_2B_ adrenoreceptors activation
Anti-depressive effects	Glutamatergic inhibition	NDMA receptor inhibition

AMPA: α-amino-3-hydroxy-5-methyl-4-isoxazolepropionic acid.

## Data Availability

Not applicable.

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
