# Peer review of "Nitrous Oxide Abuse: Clinical Outcomes, Pharmacology, Pharmacokinetics, Toxicity and Impact on Metabolism"

_toxics, 2023, doi:10.3390/toxics11120962_

Round 1

Reviewer 1 Report

Comments and Suggestions for Authors

In their review, Gent et al. described various aspects of recreational use of nitrous oxide (N2O), like its pharmacological effects, clinical manifestations, psychiatric manifestations and cardiovascular disorders (thrombosis), its effects on neurotransmitters. In addition, issues about diagnosis  are outlined.

This is a nice and rather complete review. However, it provides little added value to several reviews about N2O published before. Some important papers are missing and sometimes the authors could be more critical.

Comments

1.      The title is a bit awkward: “impact on metabolism” ! Of what? Suggest to replace this by “mechanisms of N2O-neurotoxicity”

2.      The description is a summing up of what has been described in literature, but not very critical. E.g. p. 5, line 179: “Chronic N2O use is also associated with anxiety and depression…..” I see no rationale for a causal effect. Probably, relatively more subjects with anxiety and depression tend to use N2O to relieve the symptoms. Please comment.

3.      P. 5, line 157: “Data regarding the prognosis of neurological consequences of N2O consumption is scarce.” This certainly not the case as abundant data have been published about this topic, including their treatment.

4.      3. Pharmacological impact. Suggest to replace by Pharmacological effects or Pharmacological profile.

5.      P. 6, line 205: “This short-lived psychoactive effect of N2O may precipitate frequent and heavy use. 205 Coupled to this is N2O’s reinforcing effect…” Please provide a reference for the reinforcing effect (what follows is merely a possible explanatory mechanism).

6.      P. 6, line 211. “Genetic variability to develop N2O dependence” is speculation. Please provide a reference here.

7.      Par. 3.1. I see no preclinical or clinical data about the dependence liability of N2O. I refer to older literature by Gillman et al. and a recent study by Nugteren–Van Lonkhuyzen et al. 2023.

8.      Authors should refer to reviews about N2O recently published.

9.      3.4. Anxiolytic effect. I see no human data showing an anxiolytic effect of N2O, nor of an antidepressant effect. So, both actions are speculative.

10.   Please add in 4.1. Direct N2O measurement that N2O may affect driving behaviour and may cause fatal car accidents. As such, detection is an important issue.

11.   I miss information about occupational toxicity (nurses) and use apparently preferentially by ethnical or other subgroups at increased risk.

12.   P. 10, top. I have serious doubt about the added value of Oxidative Stress markers to detect N2O use or abuse. Assay of vitamin B12, homocysteine and MMA is sufficient for diagnosis; no alternatives required.

Reviewer 2 Report

Comments and Suggestions for Authors

In this review, Gernez and colleagues discuss the nitrous oxide abuse: clinical outcomes, pharmacology, pharmacokinetics and impact on metabolism.  The authors conclude the manuscript writing that further investigations is necessary to identify other biomarkers like oxidative stress markers. The manuscript is well-written and well-organized. However, some points should be addressed.

-       The Authors should  shorten some parts of the manuscript, which are general concepts not related to the topic of this review.

-       The quality of the figures is poor. The Authors should ameliorate the quality of them.

-       Regarding other biomarkers like oxidative stress markers, the Authors should add more information. For instance, the Authors could better discuss the role of  reactive oxygen and nitrogen species in the pathophysiological mechanisms of several neurological of neuropsychiatric disorders (PMID: 25884116), and how these markers are target of several pharmacological treatments (PMID: 35002742).

-       There are several typos throughout the manuscript.

Comments on the Quality of English Language

Minor editing

Reviewer 3 Report

Comments and Suggestions for Authors

This article represents the comprehensive review of N2O, including the mechanism of action, pharmacokinetics, impact on body metabolisam, current and potential new indications and the abuse and adverse effects of the long-term use. Several review articles about specific aspects of N2O have been recently published, but this article targets more different aspects of N2O

Here are some suggestions:

At the end of the Introduction, please provide aims of this review, and the type of the review and the criteria AND methodology used to select the articles

I suggest to change 2. Pharmacological impact, to: 3. Pharmacological effects, and „Anti-depression” in table 2. to something like „Antidepressive effects” or "the effects on treatment-resistant depresion"

After brief history, the article starts with harmful long-term effects. In my opinion, the article should start with the mechanism of action, and than with two separate parts, one with approved, as well as potential N2O use for different purposes

Given several recent articles regarding the N2O potential in treatment-resistant depression, please, add some data regarding N2O in treatment-resistant depression, and also recommendation for future studies

Round 2

Reviewer 3 Report

Comments and Suggestions for Authors

I still do not see the type of the review and the criteria AND methodology used to select the articles, which was recommended. I also suggested that the article starts with the mechanism of action, followed by clinical effects, which is usually the case. The authors did not comply, because of the Reader's interest".

I also don't see additional data regarding regarding N2O in treatment-resistant depression, and where is the new reference Nagele et al, 2021, in the text

If the editor is fine with that, I have nothing more to say
